# Depression, Loneliness, Social Support, Activities of Daily Living, and Life Satisfaction in Older Adults at High-Risk of Dementia

**DOI:** 10.3390/ijerph17249448

**Published:** 2020-12-17

**Authors:** Sunghee Kim, Kwisoon Choe, Kyoungsook Lee

**Affiliations:** Department of Nursing, Chung-Ang University, Dongjak-gu, Seoul 06974, Korea; sung1024@cau.ac.kr (S.K.); kwisoonchoe@cau.ac.kr (K.C.)

**Keywords:** dementia, depression, loneliness, activities of daily living, social support, life satisfaction

## Abstract

As the number of older adults with dementia increases, early diagnosis and intervention are crucially important. The purpose of this study was to conduct dementia screening on older adults to determine whether there are differences in depression, loneliness, social support, daily activities of living, and life satisfaction between older adults at high-risk for dementia compared with low-risk older adults. We hypothesized a negative relationship between high-risk older adults and these factors. This study also hypothesized a moderating effect for social support on the relationship between daily living activities and life satisfaction. This study used a cross-sectional design with survey data. Participants were recruited at 15 public community health centers in South Korea. A total of 609 older adults (male 208, female 401) living in the community were screened for early dementia, and 113 participants (18.9%) were assigned to the high-risk group. As hypothesized, participants in the high-risk group showed significantly more negative results in terms of activities of daily living, depression, loneliness, social support, and life satisfaction compared with participants in the low-risk group. The findings of this study provide a theoretical basis for the importance of early screening for dementia and policies for effective dementia prevention.

## 1. Introduction

The rapid increase in the elderly population worldwide has been accompanied by a concomitant increase in the incidence and prevalence of dementia within this population. In 2019, the Korean Center for Dementia reported that the number of older adults with dementia over the age of 65 years was approximately 750,000 in South Korea [1]. Dementia is a disease that leads directly to a deterioration in the quality of life of older adults [2] and places severe physical, psychological, and economic burdens on both persons with dementia and their family care givers [3]. Given the increased incidence and prevalence of dementia, and the devastating effects of the disease, it is of paramount importance that dementia be detected early and that intervention is made as soon as possible to slow the progress of the disease for the patient’s and their family care givers’ quality of life.

Early screening for cognitive impairment in older adults is pivotal, particularly once older adults or their family members notice a decline in memory or difficulty performing daily activities such as paying bills, shopping, or managing medications [4]. The dementia evaluation test generally consists of a series of assessments including patient history, physical examination, functional assessment, cognitive testing, laboratory studies, and imaging studies. An initial screening test is performed for patients with suspected dementia before these are conducted.

Above all, identifying the risk factors for dementia is as essential for delaying or preventing dementia [5] as early screening. In the literature review, depression, loneliness, and poor social support were shown to be the main risk factors. Depression is generally known as a risk factor for dementia [6,7,8,9]. Depression at any age, specifically late-life depression, was associated with dementia such as Alzheimer’s disease [6], and may be a response to cognitive impairment in older persons [7]. Loneliness and poor social networks were also known the major risk factor of dementia among older adults [5,10,11,12,13,14]. Loneliness induced by low social participation and infrequent social contact was related to the occurrence of dementia [5]. For example, loneliness increased the risk of dementia among people aged 65 years and older in China [11]. Poor social networks and social support resulted in poor social engagement, and poor social engagement increased the risk of dementia [14]. Frequency of social contacts for all types of dementia was a significant protective factor [6]. In addition, social support through community engagement and marriage is known to be a protective factor against the risk of dementia [13,14]. In particular, social support from family members such as children reduced the risk of dementia, whereas negative social support increased the risk [15]. 

In addition, older adults with dementia have difficulty performing daily activities, so-called activities of daily living (ADL), across the stage of dementia [16]. ADL disability can predict the occurrence and degree of dementia [17]. Furthermore, ADL is closely related to life satisfaction of older adults with dementia [18]. Life satisfaction is a protective factor for dementia [19], and ADL affects life satisfaction among older adults regardless of dementia [20]. 

To confirm the risk factors of dementia and the usefulness of the early screening test for dementia among older adults, the research question in this study was, “does the high-risk group who was detected by early screening for dementia show any differences in variables known as risk factors for dementia when comparing with those of low-risk group?” and "is there a difference in daily living activities and life satisfaction between the high-risk and low-risk groups classified by the early screening test?". The third research question was, "Does social support have a moderating effect on the relationship between ADL and life satisfaction?". According to the conceptual model of quality of life [21], social functioning moderates the relationship between physical functioning and life satisfaction. The authors wanted to confirm whether this model applies to the group of dementia.

The purpose of this study was to identify whether there are differences in depression, loneliness, social support, daily activities of living, and life satisfaction between older adults at high-risk for dementia compared with low-risk older adults by means of an early screening test for dementia in community. Furthermore, the authors sought to identify the moderating effect of social support on the relationship between ADL and life satisfaction of older adults in both groups. 

## 2. Materials and Methods 

### 2.1. Research Design

This study used a cross-sectional design with survey data. 

### 2.2. Participants

Participants were recruited at 15 public health centers in South Korea. Public health centers are medical facilities operated by the government to promote health and prevent people’s diseases in the community. Public health centers in South Korea recommend that people in their 60s receive an early screening test for dementia, so the authors recruited those aged 60 and over who visit public health centers in their community via convenience sampling. The inclusion criteria were: older adults aged 60 years or older, willingness to take a test for early dementia screening, living in the community, and able to communicate without cognitive impairment. Older adults were not eligible if they could not communicate with discernible cognitive impairment, could not read or understand the questionnaires of this study, and did not voluntarily agree to participate in this study.

### 2.3. Instruments

#### 2.3.1. Mini-Mental State Examination for Dementia Screening (MMSE-DS) 

The Korean version of the MMSE-DS [22] was used to identify cognitive impairment in participants. The high risk for dementia was determined according to the standardized criteria (Table 1) of the previous study [22], which verified the reliability and validity of MMSE-DS. Therefore, we did not verify the reliability and validity of the scale separately in this study. This scale is commonly used as a screening test for dementia in public health centers in South Korea. MMSE-DS is composed of items that investigate orientation (time, place), memory (memory registration and recall), attention and calculation capabilities, language skills, understanding, and judgment. 

#### 2.3.2. Geriatric Depression Scale Short Form-Korea

The Geriatric Depression Scale Short Form-Korea (GDSSF-K) was used to assess depression. The GDSSF-K was standardized by Kee [23]. It comprises 15 items with a “yes” or “no” response resulting in a total possible score of 15 points. The higher the score, the higher the degree of depression. This scale is a shortened form of the 30-item Geriatric Depression Scale (GDS) developed by Yesavage et al. [24]. 

#### 2.3.3. UCLA Loneliness Scale

The revised UCLA (University of California, Los Angeles) Loneliness Scale [25] was used to measure loneliness. This is a 4-point Likert-type scale, consisting of 20 items, and the range of total score is 20–80. The higher the score, the greater the degree of loneliness experienced. 

#### 2.3.4. Instrumental Activities of Daily Living Scale 

The Korean version of the Instrumental Activities of Daily Living Scale [26] was used to evaluate everyday functions such as shopping, food preparation, housekeeping, laundry, transportation, responsibility for own medications, and handling finances. This 15-item scale was developed based on Lawton’s Instrumental Activities of Daily Living Scale [27] and rated on a 4-point Likert-type scale. The range of total score is between 15 and 60. 

#### 2.3.5. Lubben Social Network Scale 

Social support was measured using the Lubben Social Network Scale [28]. The scale measures the degree of social support received from the older person’s family network, network of friends, caregivers, and living arrangements. It consists of 10 items rated on a 6-point Likert-type scale from 0 to 5; a score of 20 or less shows a limited social network. 

#### 2.3.6. Life Satisfaction Scale

The Life Satisfaction Scale was developed by Yoon [29] based on the Memorial University of Newfoundland of Scale for Happiness [30]. It consists of a total of 20 items rated on a 4-point Likert-type scale ranging from 0 to 3 points. The higher the score, the higher the level of life satisfaction. 

### 2.4. Data Collection and Ethical Considerations

This study was conducted in accordance with the Declaration of Helsinki, and the protocol was approved by the institutional review board of Chung-Ang University (1041078-201410-HR-151-01). The authors recruited nursing and social welfare college students on a part-time basis. We trained them on the purpose of the study and how to collect the data using the questionnaire. After obtaining cooperation from the public health centers for data collection, trained college students explained the aims of the research to older persons who came for dementia screening. Participants who volunteered for this study were asked to complete a consent form and a questionnaire. The questionnaire took an average of approximately 20 min to complete, and it did not need break time in answering the questionnaires. When participants asked for help filling out the questionnaire, nursing students, social welfare students, and health center staff trained in this study helped them. After removing 11 questionnaires due to missing data, a total of 609 questionnaires were used in the data analysis.

### 2.5. Data Analysis

The data were analyzed using SPSS Statistics 23.0 (IBM Corp., Armonk, NY, USA) [31]. We analyzed the general characteristics of the participants in terms of frequencies, percentages, means, and standard deviations. As a result of the Kolmogorov-Smirnov and Shapiro-Wilk tests, the Mann–Whitney U test was used to compare differences between the two groups because the dependent variable was not normally distributed. The correlations between variables were analyzed using Spearman’s correlation analysis. Next, the authors used the standard method of determining whether a moderating effect exists, which entails adding a (linear) interaction term in a multiple regression model [32]. The four demographic variables (i.e., sex, age, education level, and marital status), ADL, and social support were added into the model step by step sequentially. In the first model, sex, age, education level, and marital status were entered. In the second model, ADL was added into the first model, and social support was added in the third model. Finally, the interaction term (ADL and social support) was added in the fourth model to test the moderating effect of social support. All statistical tests were set at the 0.05 significance level. Internal consistency reliability was assessed using Cronbach’s alpha.

## 3. Results

### 3.1. General Characteristics

A total of 609 older persons living in the community (208 males, 401 females) were screened for early dementia. Based on MMSE-DS scoring criteria, 113 people (18.9%) were assigned to a high-risk group for dementia. There were no statistically significant differences in other general characteristics except marital status (Table 2). There were more single participants (widowed, separated, divorced, unmarried) in the high-risk group.

### 3.2. Scale Characteristics

Based on the entire study sample (N = 609), the mean score for the Geriatric Depression Scale Short Form-Korea (GDSSF-K) was 4.57 (SD = 4.03). The mean score for the UCLA Loneliness Scale was 38.35 (SD = 10.60), the mean score for the Instrumental Activities of Daily Living Scale (IADL) was 52.71 (SD = 9.76), and the mean score for the Lubben Social Network Scale was 25.49 (SD = 10.09). The mean score for the Life Satisfaction Scale was 44.77 (SD = 8.03). The Cronbach’s alpha reliability for the GDSSF-K was 0.97, for the UCLA Loneliness Scale it was 0.92, and for IADL, it was 0.97. For the Lubben Social Network Scale, it was found to be 0.86. The Cronbach’s alpha of the Life Satisfaction Scale was 0.90.

### 3.3. Depression, Loneliness, ADL, Social Support, and Life Satisfaction between the Two Groups

There were differences in depression, loneliness, ADL, social support, and life satisfaction between the groups. The mean depression and loneliness scores of the high-risk group were higher than those of the low-risk group, while the mean scores of ADL, social support, and life satisfaction were lower than those of the low-risk group. This study’s effect size was small according to r, a common effect size statistic for the Mann–Whitney test (Table 3). 

### 3.4. Correlations between Variables

Positive correlations were found between depression and loneliness (r = 0.536, *p* < 0.01) and between ADLs, social support, and life satisfaction. However, negative correlations were found between two negative emotions (depression and loneliness) and positive variables (ADL, social support, and life satisfaction; Table 4).

### 3.5. Moderating Effect of Social Support on the Relationship between ADL and Life Satisfaction

As hypothesized, social support moderated the relationship between daily life activities and life satisfaction in both groups. The interaction effect of social support on the relationship between ADL and life satisfaction was statistically significant (*β* = 0.108, *p* = 0.006). The explanatory power was 27.2% in model 4. Therefore, social support has a moderating role in the relationship between ADL and life satisfaction. In both the high-risk and low-risk groups for dementia, ADL influenced life satisfaction, and the higher the social support, the higher the life satisfaction (Table 5).

## 4. Discussion

This study screened older adults living in local communities for their dementia risk using the validated Korean version of the Mini-Mental State Examination for Dementia Screening (MMSE-DS) [22]. Based on the results of this test, 113 (18.9%) participants were identified as the high-risk group, and the MMSE-DS score was lower in older adults without a spouse than the counterpart. This finding is consistent with that of a previous study [33] that older adults living alone exhibited lower MMSE-DS scores than those who did not live alone.

In this study, older adults at high risk for dementia showed significantly more negative results than those of the low-risk group. In other words, the older adults in the high-risk group experienced higher levels of depression and loneliness, and lower levels of daily living activities, social support, and life satisfaction than their low-risk counterparts. 

The finding that older adults in the high-risk group had higher depression scores than those in the low-risk is consistent with previous findings that depression increases the risk of dementia [8,9]. Depression in older adults manifests in physical symptoms such as loss of appetite and sleep disturbance, and psychological aspects such as cognitive decline and lack of motivation. Depression and dementia influence each other in diagnosis and treatment [34]. To prevent dementia in older adults, staff working at facilities for older adults in the community should provide them with interventions aimed at relieving depression. Staff working with older adults must learn to recognize depressive symptoms such as loss of appetite, lessened speech, decreased activity, and sleep disturbances for early detection of dementia. 

In this study, the high-risk group had a higher loneliness score than the low-risk group. Loneliness occurs when individuals cannot satisfy their needs in meaningful social relationships [35]. Several factors, such as deterioration in health, changes to the family structure, loss of roles, and reduced contact with family and friends can induce loneliness. Feelings of loneliness may signal early dementia [36]. Although lowered cognitive functioning may not exacerbate loneliness [37], we need to pay attention to loneliness among older adults and attempt to alleviate their loneliness to prevent dementia. 

The results of this study show that older adults in the high-risk group had lower ADL scores than those in the low-risk group. The problems of daily activities in older adults with dementia deteriorates approximately three times faster than in older adults who do not have dementia [38]. Previous studies [39,40,41] reported that a low ADL score effectively predicts the onset of cognitive disorders such as Alzheimer’s disease. Even though the ability to perform daily activities gradually decreases due to aging, we should continuously observe older adults’ ability to perform daily activities since a decline in daily activities may reflect a decline in cognitive function. 

Older adults in the high-risk group showed smaller social support network size and satisfaction than their counterparts in this study. Low social participation, infrequent social contacts, and loneliness are significantly associated with dementia [5]. In addition, this study’s results are consistent with other studies that report that social isolation is a more significant risk factor for dementia than physical inactivity, hypertension, diabetes, and obesity [42]. Conversely, Social contact is a significant protective factor for dementia [6]. Thus, it needs to promote social ties and reduce social isolation and disengagement for dementia prevention [14]. 

Moreover, the finding of this study revealed that social support has a moderating role between daily living activities and life satisfaction in both high-risk and low-risk groups in this study. This finding shows that the conceptual model of quality of life [21] can be applied to older adults with dementia as well as older adults with stroke. This result confirms once again that the importance of social support for older adults cannot be overemphasized. 

It will be possible to increase older adults’ life satisfaction by enhancing their social networks, such as participation in social activities. In the results of a study on older adults in Spain [43], perceived social support was a predictor of life satisfaction. Social support is critical for improving patients’ health outcomes. Social support is a protective factor of dementia [6,13,14,15] even in daily life activities as well as positive effects against negative emotions such as loneliness or depression. 

In this study, more older adults in the high-risk group were without spouses (widowed, separated, divorced, unmarried) than those in the low-risk group. Among other social factors, community engagement was protective for women, while for men, being married was associated with a lower incidence of dementia [13]. More research is needed to understand the relationship between dementia and marriage, but social support from a spouse may be buffering against dementia. Older adults living alone are at risk for early and late dementia [44]. Establishing meaningful social support from interpersonal relationships is a valuable preventive intervention to enhance life satisfaction and reduce the risk of dementia among older adults. In addition to family relations, there is a need to prepare various social supports in the form of relationships that seniors care for older adults with early dementia or young adults take care of older adults by residing in an intimate living space. 

Life satisfaction is an essential concept in dementia research [45,46]. In this study, the high-risk group had a lower life satisfaction score than the low-risk group. Low life satisfaction alone did not directly affect dementia but may lead to an increased death risk after five years [47]. We should constantly seek ways to increase the life satisfaction of older adults to prevent dementia. 

This study has several limitations. Participants were limited to a small sample of older adults living in a particular region of South Korea, therefore limiting the generalizability of our findings. Given that chronic physical illnesses such as diabetes, cancer, and metabolic disorders affect participants’ cognitive function, the authors should have considered these variables in this study, but not. The authors also should have checked for depression because depression can affect cognitive function but may not. These variables need to be considered in future studies on dementia. In addition, we did not take into account the male to female ratio in this study. Thus, it is necessary to consider the male to female ratio in future research. In this study, the ratio of the high-risk group and the low-risk group could not be predicted because both groups were classified according to the MMSE-DS criteria among the participants participating in the early dementia screening test. In the future, research is needed to equalize the ratio of the high-risk group and the low-risk group and compare the two groups’ variables. Another limitation derives from the study’s cross-sectional design which means that while we observed a relationship between social structure and life satisfaction, causality was not established.

## 5. Conclusions

The most important finding is that depression, loneliness, social support, ADL, and life satisfaction showed negative results in the high-risk group of dementia. Social support has a moderating role between ADL and life satisfaction in both dementia high-risk and low-risk groups. This study’s findings also provide a theoretical basis for the importance of early screening for dementia and implications for caregivers and medical personnel.

For early detection of dementia, it is necessary to carefully assess older adults’ psychological changes, such as depression and loneliness, provide psychosocial interventions for their psychological well-being, and promote social support from their social networks. It is also necessary to actively provide opportunities for older adults to belong to families and communities and make it as convenient as possible to access such resources. 

Furthermore, people should be continuously informed of the importance of early dementia screening so that older adults can receive regular early dementia screening. Above all, it is necessary to support a continuous dementia prevention program for older adults in high-risk groups and evaluate the program’s effectiveness. It is also necessary to identify differences according to the type of social support sources such as friends, co-workers, or family members separately in future studies and the types of social support services (e.g., face-to-face services and online services).

## Figures and Tables

**Table 1 ijerph-17-09448-t001:** Mini-Mental State Examination for Dementia Screening (MMSE-DS) criteria.

Age	Sex	Educational Year			
		0–3	4–6	7–12	13 and Over
60–69	Male	20	24	25	26
	Female	19	23	25	26
70–74	Male	21	23	25	26
	Female	18	21	25	26
75–79	Male	20	22	25	25
	Female	17	21	24	26
80 and over	Male	18	22	24	25
	Female	16	20	24	27

If the score is lower than that given in the table above, refer it to the diagnostic test.

**Table 2 ijerph-17-09448-t002:** Demographic characteristics.

Demographic Characteristics		High-Risk Group (*n* = 113) N (%)	Low-Risk Group (*n* = 496) N (%)	Chi-Square	
				*χ* ^2^	*p*
Sex	Male	45 (39.8%)	163 (32.9%)	1.98	0.159
	Female	68 (60.2%)	333 (67.1%)		
Marital status	Single	70 (61.9%)	236 (47.6%)	7.60	0.006
	Married	43 (38.1%)	260 (52.4%)		
Age (years)	60–69	13 (11.5%)	59 (11.9%)	1.93	0.588
	70–74	28 (24.8%)	144 (29.0%)		
	75–79	32 (28.3%)	149 (30.1%)		
	80 and over	40 (35.4%)	144 (29.0%)		
Education level	None	40 (35.4%)	216 (43.5%)	8.40	0.078
	Elementary school	45 (39.8%)	177 (35.7%)		
	Middle school	20 (17.7%)	49 (9.9%)		
	High school	5 (4.4%)	39 (7.9%)		
	Bachelor’s and higher	3 (2.7%)	15 (3.0%)		

**Table 3 ijerph-17-09448-t003:** Univariate analyses in both group (N = 609).

	Group	N	Mean Rank	U	Z	*p*	r
Depression	High-risk	113	347.60	23,210.000	−2.872	0.004	0.12
	Low-risk	496	295.29				
Loneliness	High-risk	113	364.68	20,828.000	−4.162	<0.001	0.17
	Low-risk	492	288.83				
Activities of Daily Living (ADL)	High-risk	113	266.45	23,668.000	−2.656	0.008	0.11
	Low-risk	496	313.78				
Social support	High-risk	113	253.46	22,199.500	−3.398	0.001	0.14
	Low-risk	494	315.56				
Life satisfaction	High-risk	112	247.25	21,364.500	−3.742	<0.001	0.15
	Low-risk	493	315.66				

**Table 4 ijerph-17-09448-t004:** Spearman’s correlation between variables (N = 609).

Variables	1	2	3	4	5
1. Depression	1				
2. Loneliness	0.536 **	1			
3. Activities of Daily Living (ADL)	−0.298 **	−0.297 **	1		
4. Social support	−0.325 **	−0.564 **	0.255 **	1	
5. Life satisfaction	−0.693 **	−0.632 **	0.301 **	0.438 **	1

** *p* < 0.01.

**Table 5 ijerph-17-09448-t005:** Moderating effect of social support on the relationship between ADL and life satisfaction (N = 609).

Variables	Model 1		Model 2		Model 3		Model 4	
	*β*	*p*-Value	*β*	*p*-Value	*β*	*p*-Value	*β*	*p*-Value
Sex	0.004	0.934	0.028	0.540	0.005	0.901	<0.001	0.997
Age	−0.048	0.263	0.017	0.689	0.021	0.580	0.024	0.526
Education level	0.092	0.047	0.075	0.093	0.097	0.018	0.093	0.022
Marital status	−0.118	0.040	−0.111	0.046	−0.107	0.035	−0.105	0.037
Activities of Daily Living (ADL)			0.278	<0.001	0.176	<0.001	0.222	<0.001
Social support					0.390	<0.001	0.393	<0.001
ADL × Social support							0.108	0.006
*F*	0.873	0.457	45.577	<0.001	39.180	<0.001	32.667	<0.001
*R* ^2^	0.053		0.124		0.263		0.272	
Adjusted *R*^2^	0.043		0.114		0.253		0.261	
∆*R*^2^	0.053	<0.001	0.071	<0.001	0.139	<0.001	0.009	0.006

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
