# Peer review of "Depression, Loneliness, Social Support, Activities of Daily Living, and Life Satisfaction in Older Adults at High-Risk of Dementia"

_ijerph, 2020, doi:10.3390/ijerph17249448_

Round 1

Reviewer 1 Report

Since not extensive editing has been performed compared with the previous version of the manuscript already submitted with a final decision, I cannot suggest this manuscript for a possible publication.

Thus, for compliance with my previous decision, I cannot suggest this manuscript for a possible pubblication. 

Author Response

Since not extensive editing has been performed compared with the previous version of the manuscript already submitted with a final decision, I cannot suggest this manuscript for a possible publication. Thus, for compliance with my previous decision, I cannot suggest this manuscript for a possible publication.

--> We have revised the introduction structure for internal consistency by adding some references. In particular, we provided the conceptual framework of a moderating role of social support in the relationship between physical function such as ADL and life satisfaction(on lines 70-74 on page 2). In the participant session, we asked the gender, age, marital status, and educational level (in Table 2). Still, we did not ask about general health conditions (e.g., chronic diseases, cancer, other psychotic conditions, etc.). So, we described this in the study's limitations and made future research suggestions (on lines 260-264, page 7).

We have revised the entire manuscript, including the conclusion session, to improve this paper's quality. Thank you for your review and valuable comments. Please accept this manuscript with interest.

Reviewer 2 Report

First of all, I would like to say that I am very thankful to have the opportunity to read this study. The suggestions given in this document are intended to improve your work. If you do not agree with any of them, please explain them to me, and we will try to reach a consensus. I have not received any letter from the authors responding to my previous review.

Introduction section:

Although the authors have improved the introduction section, including important information and the objective of their study, I think it still lacks structure and internal coherence. I think it can be complex for readers to read and understand.

Methods section:

  • Participants and ethics:
    • Participants information should be wider (sex, year, others, health data…)
    • What about the ethics code for the study? This is a major issue.
  • Instruments:
    • Provide psychometric information about the instruments taken from their validation and reliability studies.
    • Authors provide the reliability of the instruments based on their study. This is a result and it should be interesting in the results section and in the discussion to compare with original data.

Statistics and Results section:

  • From lines 136-144 there a lot of information which need references.
  • Please explain better how the moderating effect of Social Support is obtained, both in the methodology sections, and its interpretation in the results section.

Discussion section:

  • In this section, in addition to giving reasons why it is believed that such results have been obtained, they should be compared with other studies and trying to find the reasons why they found that results. In general, the section is poor.
  • Aren’t there future lines for this study?

Final comment:

  • The study has improved in this new submission.
  • I find a major problem in not providing an ethics commission approval number.
  • The rest of my doubts are related to a clearer and more in-depth structure of the information and data, especially in introduction and discussion sections. I find the work highly interesting, but I think it still lacks a little more depth.

Author Response

*Introduction section:

Although the authors have improved the introduction section, including important information and the objective of their study, I think it still lacks structure and internal coherence. I think it can be complex for readers to read and understand.

--> Thank you for the careful review. We have revised the introduction structure for internal consistency by adding some references (on pages 1-2). In particular, we provided the conceptual framework of a moderating role of social support in the relationship between ADL and life satisfaction(on lines 70-74 on page 2).

*Methods section:

Participants and ethics:

Participants information should be wider (sex, year, others, health data…)

-->The authors asked the gender, age, marital status, and educational level (in Table 2) but did not ask about general health conditions (e.g., chronic diseases, cancer, other psychotic conditions, etc.). So, we described this in the study's limitations and made suggestions for future research (on lines 260-264, page 7).

*What about the ethics code for the study? This is a major issue.

-->This study was approved by the Institutional Review Boards of all of the participating centers. (1041078-201410-HR-151-01)

All patients and informants provided signed informed consent to participate in the study. (on lines 134-136, 140-141)

*Instruments:

Provide psychometric information about the instruments taken from their validation and reliability studies.

-->The high risk for dementia was determined according to the standardized criteria (Table 1) of the previous study [22], which verified the reliability and validity of MMSE-DS. Therefore, we did not verify the reliability and validity of the scale separately in this study. This scale is commonly used as a screening test for dementia in public health centers in South Korea (on lines 96-99).

22. Kim, T. H.; Jhoo, J. H.; Park, J. H.; Kim, J. L.; Ryu, S. H.; Moon, S. W.; ... Lee, S. B. Korean version of mini mental status examination for dementia screening and its' short form. Psychiatry investig 2010, 7(2), 102.

*Authors provide the reliability of the instruments based on their study. This is a result and it should be interesting in the results section and in the discussion to compare with original data.

-->We described Cronbach's α of each instrument in this study to show the scales' reliability (on page 3) and cited a previous study [33] using MMSE-DS in the discussion session (on page 6).

33. Kim, H.; Lee, S.; Ku, B. D.; Ham, S. G.; Park, W. S. Associated factors for cognitive impairment in the rural highly elderly. Brain and behavior 2019 9(5), e01203.

*Statistics and Results section:

From lines 136-144 there a lot of information which need references.

-->We added two references [31,32] in the data analysis session.

31. Field, A. Discovering Statistics Using IBM SPSS Statistics, 5th ed.; SAGE Publications: Thousand Oaks, CA, USA, 2017

32. Aiken, L. S.; West, S. G. Multiple regression: Testing and interpreting interactions. 1st ; SAGE Publications: Thousand Oaks, CA, USA,. 1991.

*Please explain better how the moderating effect of Social Support is obtained, both in the methodology sections, and its interpretation in the results section.

-->According to the reviewer's comment, we added statistics about social support's moderating effect in the data analysis session (on lines 154-158). "The four demographic variables (i.e., sex, age, education level, and marital status), ADL, social support were added into the model step by step sequentially. In the first model, sex, age, education level, and marital status were entered. In the second model, ADL was added into the first model, and social support was added in the third model. Finally, the interaction term (ADL and social support) was added in the fourth model to test the moderating effect of social support".

"The interaction effect of social support on the relationship between ADL and life satisfaction was statistically significant (β = 0.108, p = .006). The explanatory power was 27.2% in model 4. Therefore, social support has a moderating role in the relationship between ADL and life satisfaction." in the results(on lines 185-188).

*Discussion section:

In this section, in addition to giving reasons why it is believed that such results have been obtained, they should be compared with other studies and trying to find the reasons why they found that results. In general, the section is poor.

-->We revised the discussion session to improve the quality of this paper. The added contents were highlighted. (on page 6-7)

*Aren’t there future lines for this study?

-->We have revised the entire conclusion session to including future lines for the study (on lines 273-288, page 8).

*Final comment:

The study has improved in this new submission.

I find a major problem in not providing an ethics commission approval number.

The rest of my doubts are related to a clearer and more in-depth structure of the information and data, especially in introduction and discussion sections. I find the work highly interesting, but I think it still lacks a little more depth.

-->According to the reviewer's comments, we have revised the manuscript as much as possible. Thanks for the specific reviewer's comment. The quality of the paper has improved thanks to reviewer comments.

Round 2

Reviewer 1 Report

The revised version of the manuscript has been positively corrected and now I can suggest the manuscript for a possible publication. 

Author Response

Thank you so much for your great review. Thanks to your review, this paper has been improved.

Reviewer 2 Report

First of all, I would like to say that I am very thankful to have the opportunity to read this study. The suggestions given in this document are intended to improve your work. If you do not agree with any of them, please explain them to me, and we will try to reach a consensus.

Methods section:

  • Again, I think that if authors provide psychometric information of the instruments based on their study, they are RESULTS.

Final comment:

  • The study has improved in this new submission.

Author Response

Methods section:

  • Again, I think that if authors provide psychometric information of the instruments based on their study, they are RESULTS.

--> As the reviewer commented, we added the scale characteristics in the results session as follows (on lines 167-174, page 5).

3.2. Scale Characteristics

Based on the entire participants (N = 609), the mean for the Geriatric Depression Scale Short Form-Korea (GDSSF-K) was 4.57 (SD = 4.03). The mean for the UCLA Loneliness Scale was 38.35 (SD = 10.60), the mean for the Instrumental Activities of Daily Living Scale (IADL)was 52.71 (SD = 9.76), and the mean for the Lubben Social Network Scale was 25.49 (SD = 10.09). The mean for the Life Satisfaction Scale was 44.77 (SD = 8.03). The Cronbach’s alpha reliability for the GDSSF-K was 0.97, for the UCLA Loneliness Scale, it was 0.92, and for IADL, it was 0.97. For the Lubben Social Network Scale, it was found to be 0.86. Cronbach’s alpha of the Life Satisfaction Scale was 0.90.

Thank you so much for taking precious time to review our paper. Thanks to you, the quality of the paper has improved. Thank you.